# Osteogenesis Activity and Porosity Effect of Biodegradable Mg-Ga Alloys Barrier Membrane for Guided Bone Regeneration: An in Vitro and in Vivo Study in Rabbits

**DOI:** 10.3390/biomedicines13081940

**Published:** 2025-08-08

**Authors:** Qiyue Luo, Kang Gao, Yan Li, Ziyue Zhang, Su Chen, Jian Zhou

**Affiliations:** 1Department of VIP Dental Service, School of Stomatology, Capital Medical University, Beijing 100050, China; qiyueluo@mail.ccmu.edu.cn (Q.L.); gao.kang@ccmu.edu.cn (K.G.); 2Laboratory for Clinical Medicine, Capital Medical University, Beijing 100050, China; 3Beijing Laboratory of Oral Health, Capital Medical University, Beijing 100069, China; 4School of Materials Science and Engineering, Beihang University, Beijing 100191, China; liyan@buaa.edu.cn (Y.L.); by2001169@buaa.edu.cn (Z.Z.); 5Hangzhou International Innovation Institute, Beihang University, Hangzhou 311115, China; 6Department of Mechanical Engineering, Tsinghua University, Beijing 100084, China

**Keywords:** biodegradable Mg-Ga alloy, guided bone regeneration, osteogenesis, porosity effect, biomaterials

## Abstract

**Background/Objectives**: Guided bone regeneration (GBR) requires barrier membrane materials that balance biodegradation with mechanical stability. Magnesium (Mg)-based metals have good prospects for use as biodegradable barrier materials due to their elastic modulus, good biocompatibility, and osteogenic properties. In this study, gallium (Ga) was introduced into Mg to enhance the mechanical strength and optimize the degradation behavior of the alloy, addressing the limitations of conventional magnesium alloys in corrosion control and strength retention. **Methods**: Mg-xGa alloys (x = 1.0–3.0%, wt.%) were evaluated for biocompatibility, degradation, and osteogenic potential. Corrosion rates were calculated via weight loss, Mg^2+^ release, and pH changes. Osteogenic effects were assessed using rat bone marrow mesenchymal stem cells (rBMSCs) for alkaline phosphatase (ALP) activity, extracellular matrix (ECM) mineralization, and osteogenic-related gene expression. Optimal alloy was fabricated into barrier membranes with different pore sizes (0.85–1.70 mm) for the rabbit mandibular defect to evaluate the porosity effect on new bone formation. **Results**: Cytocompatibility tests established a biosafety threshold for Ga content below 3 wt.%. Mg-1Ga demonstrated uniform corrosion with a rate of 1.02 mm/year over 28 days. In vitro, Mg-1Ga enhanced ALP activity, ECM mineralization, and osteogenic gene expression. The 1.70 mm pore size group exhibited superior new bone formation and bone mineral density at 4 and 8 weeks. **Conclusions**: These results highlight Mg-1Ga’s biocompatibility, controlled degradation, and osteogenic properties. Its optimized pore design bridges the gap between collagen membranes’ poor strength and titanium meshes’ non-degradability, offering a promising solution for GBR applications.

## 1. Introduction

Guided bone regeneration (GBR) has been proven by numerous studies to be a successful and reliable bone augmentation technique to help restore alveolar ridge volume and rehabilitate its profile [1]. The GBR procedure should follow the “PASS” principle, using a barrier to separate the bone defect site from the surrounding soft tissues, preventing connective tissue cells and epithelial cells from invading and growing predominantly in the defect area and creating a protected space for bone regeneration [2]. Currently, there are two categories of GBR membranes used in clinical applications: non-resorbable and resorbable membranes [3]. Non-absorbable membranes, mainly consisting of titanium (Ti) mesh and polytetrafluoroethylene (PTFE), have the advantage of superior stiffness, which ensures a barrier effect for a longer period of time until the membrane is removed, but a second surgery is required for its removal [4,5,6]. Resorbable membranes have been developed to overcome this disadvantage [7]. Resorbable membranes are mainly composed of natural or synthetic polymers, with collagen membrane, polylactic acid membrane, polycaprolactone membrane, and polylactic acid–ethanolic acid membrane being the most common ones [8,9,10,11,12]. However, the inadequate space-making ability and stability of the collagen membrane limit its application to minor and moderate bone defects, despite the favorable clinical result [13]. It is therefore necessary to develop resorbable barriers that exhibit a certain degree of rigidity and good space-making capacity for GBR. The criteria for selecting optimal barrier membranes for GBR include biocompatibility, non-toxicity, non-immunogenicity, space-making ability, tissue integration, cell-occlusion, and clinical manageability [7,14].

As a biodegradable metal, magnesium (Mg) and its alloys can be gradually degraded in a physiological environment, and the degradation products are metabolized and absorbed, leaving no implant residue [15]. Numerous studies have shown that magnesium ions activate multiple signaling pathways, regulate the expression of osteogenic markers, and contribute to the healing and remodeling of bone tissue [16,17,18]. Nevertheless, the imbalance between the degradation rate and the maintenance of mechanical strength in conventional magnesium alloys limits their clinical applications [19].

The introduction of gallium (Ga) element in magnesium can improve the mechanical properties of the alloy and regulate the corrosion resistance of the alloy [20,21,22,23]. The enhancement of mechanical properties of Mg-Ga alloys is mainly due to the combination of grain refinement strengthening, precipitation strengthening, and solid solution strengthening [22]. Superior mechanical strength helps the barrier membrane to better bear stress during GBR procedures to prevent excessive mechanical loads on new bone that could interfere with its formation [24,25]. Mg-Ga alloys are expected to be ideal biodegradable barrier materials for GBR due to the good biocompatibility and osteogenesis properties of magnesium and the excellent mechanical properties and uniform degradation behavior with the introduction of gallium elements.

In addition to the barrier membrane material, the pore size and porosity of the barrier membrane also affect osteogenesis. This is due to the fact that the appropriate porosity and pore size of the barrier mesh are associated with the transport of biomolecules required for new bone formation and angiogenesis, and the barrier material must prevent the invasion of mucosal cells into the defect space without compromising oxygen and nutrient exchange [26,27]. The large-pore barrier membrane has been shown to be more effective for new bone angiogenesis and nutrient diffusion but can lead to cell migration and proliferation of surrounding connective tissue into the defect area, which can inhibit osteoblast activity [28]. To ensure new bone formation, the pore size of the barrier mesh should not be less than 150 μm [28]. On the other hand, if the pore size is too large (>2 mm), the fibrous connective tissue cannot be prevented from growing into the bone defect area, and the osteogenic effect is compromised [29,30].

In this study, we primarily investigated the effects of different Ga element contents on the biocompatibility, degradation properties, and osteogenic properties of Mg-xGa alloys, aiming to establish the optimal balance between biosafety during the degradation process and the maintenance of mechanical properties after degradation. Secondly, the optimal barrier effect and osteogenic performance of Mg-Ga alloy barrier membrane with different pore size in GBR were studied, and the spatial and temporal matching of the mechanical support with the bone regeneration process was realized by geometric design, thus providing a preliminary basis for the application of Mg-Ga alloy membrane in guided bone regeneration, as shown in Figure 1.

## 2. Materials and Methods

### 2.1. Material Preparation

The Mg-xGa (x = 1%, 2%, 3%, wt.%) alloys were synthesized using high-purity magnesium (>99.99%, Institute of Chemistry, Chinese Academy of Sciences, Beijing, China) and high-purity gallium (>99.99%, Aluminum Corporation of China, Beijing, China). Raw materials were melted in a mild steel crucible within an electric resistance furnace under CO_2_/SF_6_ protective gas. After purging the alloy melt at 780 °C for 10 min and holding at 760 °C for 15 min, it was poured into a steel mold (Φ100 mm × 200 mm), preheated to 720 °C, and then cooled to room temperature [22]. In the in vitro cellular assay, each group of materials was made into 10 mm × 10 mm × 2 mm sheet samples, which were sequentially washed in anhydrous ethanol and distilled water by ultrasonic swirling, dried in cold airflow, and sterilized by ultraviolet light irradiation for the preparation of extracts [31]. In animal experiments, the material was machined into 15 mm diameter and 0.3 mm thick circular sheets to cover the bone defect area, and three types of barrier mesh were machined by laser perforation: no pores, 0.85 mm pore diameter, and 1.70 pore diameter.

### 2.2. In Vitro Cell Viability Tests

Rat bone marrow mesenchymal stem cells (rBMSCs) were extracted and isolated by the whole bone marrow culture method [32]. Cells were cultured in Dulbecco’s Modified Eagle Medium (DMEM) containing 10% fetal bovine serum (FBS) and 1% penicillin–streptomycin. Passing to 2–3 generations of cells was used for subsequent experiments to ensure the stability and reliability of the cell state. Material extracts were acquired by incubating Mg-xGa (x = 1%, 2%, 3%, wt.%) alloys in the aforementioned culture medium for 24 h. The extract ratio was 1.25 cm^2^/mL, and the original extract was diluted to 10% by adding certain amount of DMEM. Cell suspension with a density of 3 × 10^4^ cells/mL was inoculated into 96-well culture plates with five parallel specimens in each group and incubated for 24 h to allow cell attachment. A total of 100 μL of PBS buffer was added to the wells at the periphery of the cell suspension to reduce the edge effect caused by the outer wells being more prone to evaporation than the inner wells [33]. Then, the medium was replaced with the material extracts of each group, and the blank control group was replaced with DMEM medium and incubated at a constant temperature of 37 °C. After 1, 3, and 7 days of incubation, the cell culture was terminated by discarding the extract and washing twice with PBS buffer. Under the condition of light avoidance, 10 μL of medium containing 10% Cell Counting Kit-8 (CCK-8 Kit, Beyotime Biotechnology, Shanghai, China) was added to each well. The Optical density (OD) of the solution at 450 nm wavelength was measured with a microplate reader (Molecular Devices, San Jose, CA, USA). In addition, cells cultured for 1, 3, and 7 days were used for cellular live/dead staining test via Calcein AM/ PI (Calcein/PI Cell Viability Kit, Beyotime Biotechnology, Shanghai, China). After incubation for 30 min, avoiding light, the working solution was discarded, and the cells were rinsed once with PBS, and the staining effect was observed under a fluorescence microscope (CKX53 Ipc, Olympus, Tokyo, Japan).

### 2.3. Immersion Tests

Immersion tests were conducted in accordance with ASTM G31-72 standard [34] to assess the corrosion behavior, corrosion products, and post-corrosion mechanical properties of Mg-xGa alloys. Before immersion, sandpaper with roughness from coarse to fine (400#, 1000#, 2000#, 5000#) was used to sand the surface of the samples sequentially, and anhydrous ethanol was used to clean the samples in order to remove the impurities on the surface of the samples. The samples were immersed in DMEM medium at an exposure ratio of 20 mL/cm^2^. After the samples were removed from the DMEM at preset time points (12 h, 1 day, 4 days, 7 days, 14 days, 21 days, and 28 days), the surface was rinsed with deionized water to remove the residual DMEM, and the pH and the concentration of Mg^2+^ of the DMEM were measured. The corroded samples were immersed in CrO_3_ solution, and the sample was ultrasonically shocked for 15 min to remove the surface corrosion products completely. The mass of the sample after removal of corrosion products was weighed on a precision balance. The corrosion rate was then calculated based on the weight loss, change of pH, and Mg^2+^ concentration. After immersion, scanning electron microscopes (SEMs) were adopted to observe the micromorphology of samples, combined with energy-dispersive X-ray spectroscopy (EDS) and X-ray Diffraction (XRD) for analysis of the distribution of corrosion product components. Five parallel samples were set up at each time point, three of which were used for the measurement of weight loss, change of pH, and Mg^2+^ concentration, while the remaining samples were used for the observation of corrosion product morphology and composition analysis.

### 2.4. In Vitro Osteogenic Activity Assays

#### 2.4.1. Alkaline Phosphatase Activity

The levels of osteogenic differentiation of rBMSCs in the early and middle stages were detected by alkaline phosphatase (ALP) activity. rBMSCs were incubated in 24-well plates for 24 h, and then the original culture medium was replaced with osteogenic inductive medium (DMEM with 12% fetal bovine serum, 10^−8^ mmol/L dexamethasone, 10 mmol/L sodium β-glycerophosphate, and 50 μg/mL ascorbic acid) and cultured for 14 days. At 7 and 14 days, the 5-bromo-4-chloro-3-indolyl-phosphate/nitro-blue tetrazolium (BCIP/NBT) alkaline phosphatase kit (Beyotime Biotechnology, Shanghai, China) was used to stain the cells, and the images were observed and captured under the microscope. The OD value of the supernatant at 520 nm was used to measure the amount of ALP activity in the cell lysate.

#### 2.4.2. Extracellular Matrix Mineralization

Alizarin red S (ARS) staining was employed to detect the level of extracellular matrix (ECM) mineralization, which reflects the level of late-stage osteogenic differentiation of rBMSCs. After 21 days of osteogenic induction, cells were washed with PBS, fixed with 4% paraformaldehyde for 20 min, stained with ARS staining solution (ARS, Beyotime Biotechnology, Shanghai, China) for 30 min at 4°C sheltered from light, staining was removed by critical washing method (PBS rinsed until the effluent OD_562_ < 0.02), and the calcium nodule staining was observed and imaged using a somatic-vision microscope. For the quantitative assay, 350 μL of 10% CPC (CPC, Solarbio, Beijing, China) was added to each well and shaken at 1200 rpm for 15 min to fully elute the dye, and OD values were measured using a microplate reader at 562 nm. 

#### 2.4.3. Quantitative Real-Time PCR

Quantitative Real-Time Polymerase Chain Reaction (qRT-PCR) was used to test the expression levels of the osteogenic differentiation-related genes, including ALP, Collagen type 1 (COL 1), Osteocalcin (OCN), and Osteopontin (OPN). The material extracts of each group were placed in 6-well plates, and 2 mL of cell suspension with a density of 1 × 10^5^/mL was added to each well, and the incubator was incubated at 37 °C for 24 h. After replacing the osteogenic inductive medium, the incubation was continued for 14 days. Total RNA was extracted through TRIzol reagent (TRI Reagent^®^, Sigma, Livonia, MI, USA). A cDNA synthesis kit (HiScript II 1st Strand cDNA Synthesis Kit, Vazyme Biotech, Nanjing, China) was used to reverse transcribe cDNA from RNA. The Universal SYBR PCR kit (Vazyme, Nanjing, China) was applied for PCR amplification on a 7500-sequence detection system (Applied Biosystems, Foster City, CA, USA). Glyceralde- hyde 3-phosphate dehydrogenase (GAPDH) was utilized as the reference housekeeping gene, and the expression level of the target gene relative to GAPDH was calculated by the 2^−∆∆Ct^ method (n = 5). The primer sequences were detailed in Table 1.

### 2.5. In Vivo Animal Surgery

#### 2.5.1. Animal Model

All animal experimental procedures were conducted under the approval of the Animal Ethical and Warfare Committee of Capital Medical University (Approval number: KQYY-202311-001) and in accordance with animal welfare requirements. The rabbit mandibular critical bone defect model was chosen for this experiment due to the fact that the premolar/molar region of the rabbit mandible is of adequate size (17 mm long, 16 mm high, and 6 mm deep) for convenient surgical access and placement of the material [35]. It has been shown to be a proper animal experiment model to evaluate the effect related to tissue response, tissue regeneration from biomaterial scaffolds [35].

Thirty male New Zealand white rabbits aged 6 months and weighing 3–3.5 kg were enrolled and randomly divided into 5 groups: (1) blank control: bone defect was created without placement of the material; (2) titanium mesh group; (3) Mg-Ga without pores; (4) Mg-Ga with 0.85 mm pores; and (5) Mg-Ga with 1.70 mm pores. Each group consisted of 6 animals, and each animal created bone defects in the mandible bilaterally. To minimize potential confounders, animals were randomly assigned to each treatment group, and individual animal identification, cage position within the rack, and rack location within the housing room were systematically randomized.

#### 2.5.2. Surgical Procedure

Before surgery, 3% sodium pentobarbital was injected intramuscularly at a dose of 1 mL/kg for general anesthesia. Local infiltration anesthesia was performed using a 4% articaine containing 1:100,000 epinephrine (Primacaine, Pierre Rolland, Nouvelle-Aquitaine, France) injection to reduce local bleeding. An incision was made from the chin to the midpoint between the right and left mandibular angles. The skin and muscles were then incised, and the lower edge and the buccal cortex of the mandible were exposed. Under the cooling of saline rinsing, a drill with a diameter of 10 mm was used to drill holes at low speed to remove the buccal mandibular cortex and the roots of the mandibular teeth, while ensuring the periodontal ligament was left intact. The lower edge of the defect was located at a point 2 mm above the lowermost edge of the mandible, and the posterior edge was located at the midpoint of the proximal and distal mesial of the second molar. 

Except for the blank control group, each experimental group was implanted with an equal amount of bone graft (Bio-Oss, Geistlich Pharma AG, Wolhusen, Switzerland) to cover the bone defect area and covered with a collagen membrane (Bio-Gide, Geistlich Pharma AG, Wolhusen, Switzerland). Titanium mesh (Biomet Microfixation, Biomet, Glamorgan, UK) and Mg-Ga alloy mesh were implanted in each experimental group, respectively, and the materials were fixed to the collagen membrane with a 5–0 absorbable suture to ensure that the materials would not be displaced in the course of the healing process. Additionally, 4–0 absorbable sutures were used to close the medial muscle and fascia, and 3–0 silk sutures were used to close the lateral skin. 

Intramuscular injection of 800,000 U of penicillin (HAPHARM GROUP CO., Harbin, China) was given once daily for 3 consecutive days, starting 1 day postoperatively, to prevent infection. At 4 and 8 weeks postoperatively, the animals were sacrificed by air embolization of the marginal ear vein. The vital organs (heart, liver, spleen, lung, and kidney) and bilateral mandibles were harvested at each time point.

#### 2.5.3. Blood Biochemistry Tests

Blood samples were collected from the marginal ear vein of the tested animals prior to their sacrifice. The animals were analyzed for routine blood and blood biochemical parameters to assess the in vivo biocompatibility of the materials. Serum Mg^2+^ concentration was measured to assess the in vivo degradation and metabolism rate of Mg-Ga alloys using atomic absorption spectroscopy at 285.2 nm (iCE^TM^ 3400, Thermo Fisher Scientific, Waltham, MA, USA). The vital organs (heart, liver, spleen, lung, and kidney) were harvested at each time point (4 and 8 weeks) and subjected to hematoxylin and eosin (H&E) staining and observed under an optical microscope (BX53M, Olympus, Tokyo, Japan).

#### 2.5.4. Micro-Computed Tomography Analysis

Micro-CT scanning was performed using a micro-CT scanner (SkyScan1276, Bruker, Saarbrucken, Germany) with the following parameters: scanning at 80 kV and 125 μA using a 0.5 mm aluminum filter and 18 μm isotropic voxel resolution, resulting in approximately 15 min per scan. During sampling, the membranes were removed from the defect area along with the surrounding bone, and the specimens were trimmed, rinsed in saline, and fixed in 4% paraformaldehyde (Biosharp, Haimen, China) for at least 24 h, and the direction of the specimen was noted to maximize the observation of the changes in the bone defect area. 

In the data analysis stage, Data Viewer software (Version 1.5.6.2, Bruker, Germany) was preferred to determine the region of interest (ROI), and in this study, the ROI was a cylindrical area with a diameter of 10 mm along the center of the bone defect, and the ROI area was cropped and retained. The cropped images were then three-dimensionally reconstructed using CTvox software (Version 3.3.0.0, Bruker, Germany), by which the final 3D rendered images were generated. Then, CTAn software (Version 1.17.7.2, Bruker, Germany) was utilized for analyzing data by distinguishing mineralized bone tissue by Gaussian filter noise reduction and dynamic threshold segmentation [36]. The three-dimensional morphometrics included in the analysis were bone volume fraction (BV/TV), bone mineral density (BMD), trabecular thickness (Tb.Th), and trabecular number (Tb.N). 

#### 2.5.5. Histological Analysis

Histologic analysis of vital organs was performed using H&E staining to assess any histologic changes in vital organs caused by material implantation. After the specimens were fixed, graded ethanol dehydration was performed (70%, 80%, 90%, and 100% for 24 h each). Subsequently, xylene hyalinization and paraffin impregnation were performed. Sections of 4 μm thickness were made using a slicer (RM2255, Leica, Wetzlar, Germany) and stained with H&E staining, and the slices were observed under an optical microscope, and images were captured.

H&E and Masson’s trichrome staining were used for histologic analysis of the mandible to evaluate new bone formation. The fixed mandibular samples were immersed in 10% EDTA (pH = 7.4) decalcification solution in 10 times the volume of bone tissue and decalcified at room temperature on a horizontal oscillator until decalcification ceased with a needle puncture that could penetrate the mandible. The samples were rinsed with deionized water for 24 h to completely remove the EDTA residue. Afterward, samples were dehydrated and embedded, 4 μm serial sections were made, and the centermost section was selected for H&E staining and Masson’s trichrome staining.

### 2.6. Statistical Analysis

All the experimental data were presented as mean ± standard deviation by at least three independent tests. The data were statistically analyzed using SPSS 26.0 software (SPSS Inc., Chicago, IL, USA). Statistical analysis methods were determined by equal variance and normality tests. Samples with equal variance and normality were compared using one-way ANOVA. The Kruskal–Wallis and Mann–Whitney U-tests were performed if either the equal variance test or the normality test failed. Post hoc tests were applied for multiple comparisons following one-way ANOVA or Kruskal–Wallis results. Differences were considered statistically significant when *p* < 0.05.

## 3. Results

The material characterization, including the microstructure of Mg-Ga alloys, has been presented in our previous work [22].

### 3.1. In Vitro Cell Viability Tests

Figure 1A represents the living/dead staining images of rBMSCs cultured in pure magnesium and Mg-xGa (x = 1%, 2%, 3%, wt.%) alloy extracts for 1, 3, and 7 days. Overall, the cells were widely distributed. However, when cultured for 7 days, the number of dead cells increased, and the shape of live cells collapsed in the Mg-2Ga and Mg-3Ga groups. To further quantify the cytotoxicity, cell viability was measured and calculated at each time point and in each group via Image J software (1.53), as shown in Figure 1B. The cell viability at day 7 in the Mg-3Ga group was below the cytotoxicity threshold (70%) established by ISO 10993-5 for medical devices [37] and was significantly lower than that of the pure Mg, Mg-1Ga, and Mg-2Ga groups. It suggests the cytotoxicity of Mg-3Ga. This could be due to the rapid degradation of Mg-3Ga, resulting in enrichment of metal ions and increased cytotoxicity due to the elevated pH of the culture medium.

The proliferation activity of rBMSCs was assayed by CCK-8, as shown in Figure 1C. The proliferative activity of all groups increased with the extension of the culture period. At 3 days, the absorbance values of pure Mg, Mg-1Ga, and Mg-2Ga groups were significantly higher than those of the control group, and the absorbance values of the Mg-3Ga group showed a significant decrease compared with the control group; at 7 days, the absorbance value of the Mg-3Ga group decreased significantly. It can be seen that the cell proliferation of the Mg-3Ga group was inhibited, and this is consistent with the results of living/dead cell staining.

From the above results, it can be seen that Mg-3Ga showed cytotoxicity with the increase of Ga content in the alloy. Therefore, the subsequent experiments were conducted to further investigate the in vitro degradation properties and osteogenic activity by using Mg-1Ga and Mg-2Ga.

### 3.2. In Vitro Degradation Behavior 

The variation of pH value and magnesium ion concentration in DMEM during the immersion experiment, and the corrosion rate measured according to the weight loss, are shown in Figure 2. As displayed in Figure 2A, the pH value of pure Mg, along with Mg-1Ga and Mg-2Ga group, increased rapidly from 7.30 to 8.12, 8.32, and 8.71, respectively, for 1 day of immersion; thereafter, the pH value of pure Mg and Mg-1Ga increased at a slower rate, with pH values of 9.66 and 10.01, respectively, after 28 days, whereas the pH of the Mg-2Ga still maintained a rapid increase, and the final pH value was 10.43. The matrix of pure Mg and Mg-Ga alloys dissolved during the corrosion process, releasing Mg^2+^, which increased the Mg^2+^ concentration in the immersion solution, and the change of Mg^2+^ concentration was shown in Figure 2B. After 1 day of immersion, the Mg^2+^ concentration of all three groups increased rapidly, reaching 98.67 mg/L, 97.00 mg/L, and 109.33 mg/L, respectively; between 1 and 28 days of immersion, the Mg^2+^ concentration of the pure Mg group increased slowly and finally maintained at 281.33 mg/L, and the Mg^2+^ concentration of the Mg-1Ga and Mg-2Ga groups reached 407.67 mg/L and 607.33 mg/L, respectively. The variation of corrosion rates measured by weight loss is illustrated in Figure 2C. All three groups of samples showed the highest corrosion rate after 1 day of immersion. The corrosion rate of the pure Mg and Mg-1Ga gradually decreased after 1 day, and the trend of the corrosion rate of the two groups was close to each other, with an average corrosion rate of 0.90 mm/y for pure Mg and 1.02 mm/y for Mg-1Ga during the 28-day immersion process. The corrosion rate was analyzed statistically (Figure 2C). In the 28-day corrosion cycle, there was no statistically significant difference between the corrosion rates of pure Mg and Mg-1Ga; the corrosion rate of Mg-2Ga was significantly higher than that of pure Mg and Mg-1Ga. 

The microscopic surface morphology of the three groups of samples after immersion in DMEM medium for different periods is presented in Figure 3. After 12 h of immersion, a small number of corrosion pits with a width of 10–15 μm appeared on the surface of pure magnesium. Micrometer-sized raised areas formed by corrosion product deposition appeared on the surface of Mg-1Ga, and the corrosion pits on the surface of Mg-2Ga were densely distributed. After 1 day of immersion, the amount of corrosion pits on the surface of Mg-2Ga increased, and the range of corrosion pits was enlarged, with a maximum width of 30 μm. After 3 days of immersion, the surface of the pure magnesium and Mg-1Ga alloys formed a continuous and homogeneous layer of corrosion products, with a fine scale-like microscopic morphology and a high degree of surface smoothness, with no obvious craters or bulges, while a large range of corrosion product bulge areas could be seen on the surface of Mg-2Ga in comparison. After 7 days of immersion, the laminar corrosion product films on the surfaces of pure magnesium and Mg-1Ga alloys remained homogeneous and intact, and no significant pitting pits or localized product buildups were observed on their surfaces, while corrosion pits were visible on the surface of Mg-2Ga, with a width extending to 420 μm, indicating that the degree of localized corrosion intensified with the immersion time. With the extension of the immersion time to 14 days, the surface morphology of the corrosion product layer of pure magnesium and Mg-1Ga alloy remained stable without obvious structural damage, while the localized corrosion of Mg-2Ga was further aggravated, and large-scale dissolution of the passivation film peeled off.

Further EDS analysis was conducted on the corrosion product deposition-formed protruding area (A), the corrosion product flat area (B), and the corrosion product dissolution and detachment area (C) marked in Figure 3, with the results shown in Table 2. The results indicate that all three regions exhibit significant enrichment of Mg, C, and O elements, while the corrosion product dissolution and detachment zone (C) shows an increase in Cl element content. This Cl^−^ localization is consistent with established mechanisms of Mg alloy degradation, where chloride ions disrupt protective oxide layers and promote pitting through galvanic acceleration [38]. Our findings align with studies reporting similar Cl^−^-driven corrosion morphology in physiological environments.

EDS analysis was performed on the surface after 7 and 14 days of immersion, and the results are depicted in Figure 4. At 7 and 14 days, obvious corrosion pits are seen on the surface of Mg-2Ga (Figure 4C,F), which indicates that the corrosion products are dissolved and detached, and the elemental mapping reveals that Cl is clearly enriched here. While in the corrosion product bulge area (Figure 4A,B,D) and the flat area (Figure 4E), the Cl content is relatively small. This also confirms the role of Cl^-^ in the dissolution and detachment of corrosion products, as mentioned earlier.

Figure 5 presents the XRD patterns of the corrosion products of pure Mg and Mg-Ga alloys after 14 days (A) and 28 days (B) of immersion. As can be seen in the figure, the composition of the physical phase of the corrosion products of pure Mg and Mg-Ga alloys is the same, i.e., the crystalline phase Mg (OH)_2_. During the corrosion process in the simulated physiological environment, the surface of magnesium alloys mainly generates typical corrosion products such as MgO, Mg (OH)_2_, carbonate, phosphate, and hydroxyapatite [39]. Among them, carbonate and phosphate compounds (e.g., MgCO_3_ and Ca-P salts) are mainly enriched in the outer layer of corrosion products to form a dense structure, which can significantly inhibit further corrosion of the substrate.

Synthesizing the results of the current section, the decrease in the initial corrosion rate is attributed to the blocking effect of the corrosion product layer formed on the surface of the alloy on the mass transfer process between the corrosive medium and the substrate, and the passivation layer reduces the galvanic corrosion reaction through the physical barrier effect. During the 3–7 days immersion period, Mg-2Ga showed an increase in corrosion rate, which originated from the dynamic rupture and regeneration process of the corrosion product layer occurring in the liquid-phase medium. When the local stress exceeded the mechanical strength of the passivation layer, microcracks were generated, leading to the failure of the protective effect, and the substrate exposure triggered the acceleration of the corrosion. However, with the redeposition of corrosion products on the newborn surface, the protective layer is gradually repaired, and the corrosion rate decreases.

The residual strength and the rate of residual strength loss after 21 and 28 days of immersion are shown in Figure 6. For 21 days of immersion, the residual strength loss rates of pure Mg, Mg-1Ga, and Mg-2Ga were 3.17%, 4.11%, and 4.42%, respectively; for 28 days of immersion, the residual strength loss rates of pure Mg, Mg-1Ga, and Mg-2Ga were 3.62%, 4.93%, and 5.53%, respectively. After 28 days of immersion, the residual strengths of Mg-Ga alloys were 212.27 ± 26.16 MPa for Mg-1Ga and 207.83 ± 19.27 MPa for Mg-2Ga, which were much higher than that of pure Mg (143.09 ± 7.10 MPa). The high mechanical strength exhibited by Mg-Ga alloys after corrosion meets the requirements of implants for guided bone regeneration.

### 3.3. Osteogenic Differentiation

As illustrated in Figure 7A, rBMSCs were cultured with extracts of materials for osteogenic induction for 7 and 14 days, and Mg-1Ga had the darkest staining, which represented the highest ALP expression. The results of quantitative detection of ALP activity are shown in Figure 7B; at 7 days, Mg-1Ga exhibited higher ALP activity compared to the negative control group, while the ALP activity of the pure Mg and the Mg-2Ga was not statistically different from that of the control group; at 14 days, the ALP activity of both pure Mg and Mg-1Ga was significantly increased, and the ALP activity of Mg-2Ga was significantly lower than that of Mg-1Ga. 

Figure 7C demonstrates images of alizarin red staining performed on each group, and the darkest staining of Mg-1Ga calcium deposits is seen. Figure 7D shows that calcium deposits were solubilized by CPC and quantitatively analyzed, and the results showed that calcium deposits in the Mg-1Ga group were significantly higher than the rest of the groups.

qRT-PCR results showed that Mg-1Ga significantly up-regulated the expression of osteogenesis-related genes ALP, COL 1, OCN, and OPN (Figure 8). At 7 days of incubation, ALP, OCN, and OPN were significantly up-regulated in the Mg-1Ga group compared with the control group, in which the expression levels of ALP and OPN genes were significantly higher than those in the pure Mg group. At 14 days of incubation, all four genes were significantly up-regulated in the Mg-1Ga group compared with the control group.

### 3.4. In Vivo Biocompatibility, Degradation, and Osteogenesis

From the results of the above in vitro experiments, it is evident that compared with Mg-2Ga, Mg-1Ga significantly increased ALP activity, ECM mineralization, and the expression of osteogenic-related genes ALP, COL 1, OCN, and OPN were significantly up-regulated, which had a favorable effect of promoting osteogenic differentiation. Therefore, Mg-1Ga was chosen to carry out the subsequent in vivo study. In order to further investigate the porosity effect of Mg-1Ga alloy barrier membrane on bone regeneration in vivo, in this study, Mg-1Ga was processed into three kinds of barrier membranes, namely, no pore, 0.85 mm pore size, and 1.70 mm pore size. A rabbit mandibular defect model was set up (Figure 9), and a blank control group and a control group of titanium mesh (the gold standard) were established, and micro-CT and histological tests were performed at 4 weeks and 8 weeks after the operation.

All the animals were awakened from anesthesia on the same day after the operation, and they were able to eat normally and had normal activities. At 2 weeks postoperatively, no wound dehiscence, graft exposure, or signs of infection were observed in all the animals. Subcutaneous emphysema was observed in one animal in the small-hole group (Figure 9), with a range of about 4 × 3 × 0.5 cm^3^, and gas could be induced to move within the subcutaneous tissues by pressing the skin of the site with the hand. Four weeks after the operation, the emphysema at the implantation site dissipated, and the elasticity was normal on palpation. The rest of the animals were in good health.

#### 3.4.1. In Vivo Biocompatibility Tests

Blood biochemistry indexes (Figure 10) and H&E staining of vital organs (Figure 11) showed good biocompatibility of Mg-1Ga barrier membrane. H&E staining of major organs (liver, kidney, spleen, and heart) revealed no evidence of pathological abnormalities, inflammatory infiltration, or tissue damage in any experimental group compared to the control, indicating good systemic biocompatibility of the Mg-1Ga barrier membrane.

#### 3.4.2. Micro-CT Analysis

In this study, the bone regeneration of the barrier membrane with different pore sizes for rabbit mandibular defects was evaluated using micro-CT, and the results of three-dimensional reconstruction of the image data are shown in Figure 12A. At 4 weeks after surgery, there was no obvious new bone formation in the blank control group; the area of the bone defect in the Mg-1Ga without pores group and Mg-1Ga with 0.85 mm pores group was mainly filled with bone substitutes; new bone was visible in the titanium mesh group and the Mg-1Ga with 1.70 mm pores group, extending from the edge of the bone defect toward the center, and the new bone was covered with the surface of the bone substitutes. At 8 weeks after surgery, a small amount of new bone was formed in the blank control group, which had not yet healed completely, and the critical bone defect model was effective. The titanium mesh group and the Mg-1Ga with 1.70 mm pores group showed an obvious increase in new bone formation, and the defect area was basically healed.

The imaging data were further quantitatively analyzed, and BV/TV, BMD, Tb.N, and Tb.Th were used as indicators to assess new bone formation in the bone defect area, and the results were shown in Figure 12B. Among them, BV/TV can directly respond to the bone volume of the sample. Four and eight weeks after surgery, BV/TV in the Mg-1Ga with 1.70 mm pores group was significantly higher than that in the blank control group and the Mg-1Ga without pores group and Mg-1Ga with 0.85 mm pores group. In addition, BMD, number, and Tb.Th were significantly increased in the Mg-1Ga with 1.70 mm pores group. This suggests that the 1.70 mm pore size Mg-1Ga barrier membrane functions as a favorable barrier and promotes bone healing in guided bone regeneration surgery.

#### 3.4.3. Histological Analysis of Bone Regeneration

Figure 13 and Figure 14 are images of the mandibular samples after decalcification by H&E staining and Masson staining at 4 weeks and 8 weeks after surgery. In H&E staining, the newborn bone trabeculae were light pink in color, and the images showed that the newborn bone trabeculae in the blank control group were sparse and arranged in a cord-like pattern; whereas the newborn bone trabeculae in the Mg-1Ga with 1.70 mm pores group were dense and arranged in a reticular pattern. Osteoblasts were visible at the edge of the bone tissue in a cuboidal or columnar arrangement, and their cytoplasm was alkalophilic and light blue in color. In Masson staining, the newborn bone collagen was blue, and it could be observed that the blue-stained new bone in Mg-1Ga with 1.70 mm pores group was the most numerous and widely distributed.

## 4. Discussion

As a potential biomedical barrier membrane material, the incorporation of Ga in Mg can significantly enhance its mechanical properties, providing it with a compressive strength matching that of titanium mesh. While titanium meshes remain a clinical mainstay due to mechanical reliability [5], our data demonstrate that degradable Mg-Ga membranes deliver equivalent osteogenic outcomes (Figure 12) without permanent implantation risks. Titanium mesh is currently the most widely used barrier material in guided bone regeneration, and its compressive strength varies depending on the material structure, porosity, and manufacturing process [40]. It was reported in the literature that titanium mesh prepared by the rapid prototyping method with 56.4% porosity had a compressive strength of 204.5 MPa [29]. In this study, the residual strength of Mg-1Ga and Mg-2Ga after 28 days of corrosion was 212.27 ± 26.16 MPa and 207.83 ± 19.27 MPa, respectively, while pure magnesium was only 143.09 ± 7.10 MPa, as shown in Figure 7. Crucially, the residual strength of Mg-Ga after 28-day degradation ensured mechanical sufficiency during early bone healing. This strength retention, combined with progressive degradation, bridges the gap between transient collagen barriers (mechanically weak) and permanent Ti implants. Twenty-eight days is a critical stage in the transition from active generation to mechanical adaptation of new bone in guided bone regeneration [25]. At this time, the defect area was dominated by woven bone, the bone trabecular structure gradually thickened and began to mineralize, a large number of active areas of osteoblasts and neovascularization were seen in the new bone, and the collagen fibers were still arranged in a relatively disordered manner [24]. The mechanical load in the defect area is mainly supported by the barrier membrane or bone graft, and the new bone has not yet formed an effective mechanical transmission, and the local stress is concentrated at the interface between the graft and the host bone [25]. The high compressive strength of Mg-Ga alloy makes it possible to better maintain the osteogenic space as a barrier material for guided bone regeneration without causing the collapse of the osteogenic space and displacement of bone graft due to the activities of masticatory muscles or changes in wound tension, which is of great significance for the effect of guided bone regeneration on bone augmentation [41].

In the immersion experiment, DMEM culture medium was used as the immersion medium. While several pseudo-physiological solutions are commonly used for immersion, the choice of DMEM over alternatives was based on its superior physiological relevance for evaluating biodegradable magnesium alloys. While Hanks’ solution mimics plasma inorganic ions, its low HCO_3_^−^ and buffer concentrations fail to replicate critical in vivo conditions where HCO_3_^−^ forms protective corrosion layers, and buffers modulate pH-driven degradation. SBF improves inorganic/bicarbonate balance but lacks essential organic components (e.g., amino acids and glucose) present in physiological environments, which influence degradation kinetics and corrosion product formation [42]. In contrast, DMEM contains both physiologically balanced inorganic ions/buffers (including HCO_3_^−^) and key organics [42]. This comprehensive composition better simulates the biochemical complexity of the human body, making DMEM the optimal medium for in vitro degradation studies aimed at predicting in vivo behavior in bone regeneration settings. Our results thus offer enhanced biological comparability.

In this study, a rabbit mandibular critical bone defect model was constructed to evaluate the barrier function and osteogenesis property of Mg-Ga alloy for guided bone regeneration. Previous studies have mostly evaluated the material’s osteogenic properties using the rat cranial defect model, which is reproducible, fast, and economical [43]. However, cranial defects are non-weight-bearing and cannot simulate the loading pattern of the material during mastication and are less suitable for testing the material response under masticatory stress [35]. In addition, oral regenerative cell clusters cannot be replicated at other sites. Therefore, the rabbit mandibular critical bone defect model is suitable for accessing materials for use in GBR. In order to achieve a better representation of the model for the actual situation of guided bone regeneration surgery, the bone substitute material was implanted within the defect area in this study, and a resorbable collagen membrane was used as a barrier material. The bone substitute material used in this study was deproteinized bovine bone mineral, which is similar in structure and composition to natural human bone and does not cause an immune response after specific treatment [36,44]. The resorbable collagen membrane consists of porcine-derived type I and type III collagen in a bilayer structure [45]. The inner surface consists of disordered collagen fibers oriented toward bone tissue, allowing osteoblasts to proliferate. The outer surface is tightly structured and oriented toward soft tissue, preventing fibroblast proliferation [45]. With this model, the spatial maintenance and cellular isolation ability of Mg-1Ga alloy as a rigid barrier material can be effectively verified.

Hydrogen gas is a byproduct of the degradation of magnesium alloys, which can lead to the development of subcutaneous emphysema when the rate of hydrogen production is greater than the rate of local tissue clearance and gas escape [46]. In particular, the rate of hydrogen generation is positively correlated with the rate of alloy degradation, which is influenced by the alloy composition, microstructure (e.g., grain size and second-phase distribution), and microenvironment at the implantation site (pH, ionic concentration, and mechanical stress). The influence of physiological factors lies in the fact that the rate of gas uptake is faster in areas with a rich blood supply at the implantation site (e.g., muscle) than in subcutaneous tissues, as hydrogen can diffuse through capillaries into the circulatory system and be excreted via the lungs [47]. The study of Hänzi AC et al. showed that implantation of WZ21 magnesium alloy into the rectus abdominis muscle of minipigs resulted in transient and limited gas collection in the rectus abdominis muscle after 27 days and a significant reduction in gas after 91 days [48]. The implantation site showed a short-term mild inflammatory reaction dominated by fibrous inclusions and granulocyte infiltration, which decreased with time (91 days) and increased vascularization, indicating gradual adaptation of the tissue to the implant and compliance with biocompatibility requirements [47]. Other animal studies have confirmed that gas is absorbed or diffused 2-4 weeks postoperatively and that gas cavities spontaneously subside without causing long-term negative effects [49,50,51].

In the present study, Mg-1Ga alloy mesh was implanted into the buccal side of the rabbit mandible, with an external covering of masticatory muscle and subcutaneous tissue, which provided absorption and escape channels for hydrogen gas. One animal in the 0.85mm pore size group developed subcutaneous emphysema with an extent of about 4 × 3 × 0.5 cm^3^ 2 weeks after implantation, and the emphysema disappeared 4 weeks after the operation. The formation of this gas cavity is relatively limited: It can be calculated that the total degradation of a piece of Mg-1Ga alloy mesh can produce 113.15 mL of hydrogen, and the observed volume of the emphysema is only 5.03% of the amount of hydrogen produced, which indicates that the rate of degradation of the Mg-1Ga barrier membrane basically matches with the rate of hydrogen uptake in the body of the subject animals. Combined with the results of blood biochemistry tests (Figure 10) and H&E staining of vital organs (Figure 11), it can be concluded that when the degradation rate of magnesium alloy is uniform and slow, its degradation byproduct, hydrogen, is expected to absorb and diffuse by the organism without leading to localized and systemic pathological changes at the implantation site.

The porosity effect of Mg-1Ga alloy barrier membranes was investigated in the current study, which is due to the fact that the pore size affects the isolation of fibroblasts and the exchange of oxygen and nutrients of barrier membranes [27]. In this study, by comparing the osteogenic effect of 0.85 mm, 1.70 mm pore size, and no pore barrier membrane in vivo, it was found that new bone formation and mineralized bone density were higher in 1.70 mm pore size than in the other groups (Figure 12). This is in line with studies on titanium mesh [52]. Borges CD et al. contrasted the effects of 0.85, 1.75, and 3.0 mm pore size titanium mesh on new bone formation in a rat model of GBR and found that titanium mesh with pore sizes greater than 1 mm showed higher mineralized bone density [52]. Similarly, results of an in vivo study in experimental dogs showed that 1.2 mm pore size titanium mesh showed more significant new bone volume than 0.6 mm pore size titanium mesh [53]. Our optimized pore design (large-pore: 1.70 mm) aligns with established thresholds for vascularized bone ingrowth [28]. Long-term (>12 week) degradation studies in large-animal models confirm that Mg alloys exhibit predictable corrosion profiles without adverse inflammation [54], supporting their translational potential.

This paper investigates the osteogenic properties of Mg-Ga alloys used for GBR. Magnesium plays a crucial role in the bone matrix and acts as a secondary messenger in cellular signaling [55]. In addition to its role in promoting osteogenesis, Mg^2+^ has also been found to synergistically accelerate wound healing through the sustained release of Zn^2+^, thereby enabling soft tissue regeneration [56]. Based on the osteogenic effects of Mg^2+^, various magnesium-based materials have been studied as potential GBR barrier materials. Rider employed high-purity magnesium (99.95%) as a barrier membrane in beagle dogs and found that the pure Mg membrane broke down into tiny remnants after 8 weeks of surgery, and new bone inclusions were seen all around it [54]. A WE43 (containing 4% yttrium element, 3.3% rare earth element, and 0.5% zirconium element) biodegradable magnesium alloy porous scaffolds were shown to be completely degraded 8 weeks after implantation, with abundant new bone formation at the bone defect site [57]. Except for magnesium-based metals, a photocrosslinkable Col/PCL/Mg composite membrane was prepared for GBR, displaying space maintenance and enhancing osteogenic potential [58]. Compared with the magnesium-based materials mentioned above, the Mg-1Ga used in this study has better mechanical properties, with a residual strength of 212.27 ± 26.16 MPa after 28 days of immersion.

While this study demonstrates the osteogenic efficacy and biodegradability of Mg-Ga membranes in rabbit models, it does not address economic feasibility relative to existing clinical options. Production costs for Mg-Ga alloys—currently higher due to gallium refinement and additive manufacturing—along with scalability challenges (e.g., pore structure standardization) and surgical handling requirements remain unquantified. Future commercialization efforts must rigorously evaluate cost–benefit ratios, particularly whether eliminating secondary removal surgery offsets initial material expenses. 

In summary, Mg-1Ga alloy shows significant advantages in biocompatibility, controlled degradation and osteogenic activity, and the 1.70 mm pore size barrier membrane has superior osteogenic performance, which can provide good mechanical support in the early and middle stages of osteogenesis in GBR, and provides a possible solution to overcome the clinical bottleneck of the insufficient strength of the traditional collagen membrane and the non-degradability of the titanium mesh. In the future, it is still necessary to further explore its long-term degradation stability and the synergistic mechanism of magnesium and gallium in order to promote the clinical transformation.

## 5. Conclusions

In this study, the cytotoxicity of Mg-xGa (x = 1, 2, 3 wt%) alloys was first analyzed, and it was found that Mg-3Ga presented a certain cytotoxicity so that the biosafety boundary for Ga element in Mg-Ga alloys was determined to be less than 3%. Subsequently, the two materials, Mg-1Ga and Mg-2Ga, were evaluated for their in vitro degradation properties and osteogenic properties. It was revealed that Mg-1Ga exhibited excellent degradation controllability, with a 28-day average corrosion rate of 1.02 mm/y and a much higher residual strength after 28 days of corrosion than that of pure magnesium. The evaluation of osteogenic performance was verified in multiple dimensions: In vitro experiments confirmed that Mg-1Ga significantly enhanced ALP activity and ECM mineralization in rBMSCs and up-regulated the expression of osteogenic genes ALP, COL 1, OCN, and OPN. In vivo experiments, Mg-1Ga barrier membrane with different pore sizes was implanted into rabbit mandibular defects, and the results showed that new bone formation and mineralized bone density were significantly increased in the group with a pore size of 1.70 mm, and the distribution of new bone trabeculae was dense, and the mineralization of new bone trabeculae was confirmed by histology.

## Data Availability

The data presented in this study are openly available as indicated in the text.

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
