# Peer review of "Osteogenesis Activity and Porosity Effect of Biodegradable Mg-Ga Alloys Barrier Membrane for Guided Bone Regeneration: An in Vitro and in Vivo Study in Rabbits"

_biomedicines, 2025, doi:10.3390/biomedicines13081940_

Round 1

Reviewer 1 Report

Comments and Suggestions for Authors

This study systematically evaluated the biological properties of biodegradable magnesium-gallium alloys as barrier materials for guided bone regeneration. It was demonstrated that Mg-1Ga alloy exhibits significant advantages in terms of biological safety, controllable degradation, and osteogenic activity. The 1.70 mm pore size barrier mesh showed excellent osteogenic properties and were able to provide good mechanical support in the early stage of guided bone regeneration. The idea is innovative and the manuscript is well organized and written. However, there are still some questions to be answered and some problems to be modified:

  1. A graphical abstract is needed to illustrate the research approach and content of this study more intuitively.
  2. Certain symbols in the text are written incorrectly and need to be corrected. For example, in page 4, line 28, “10*10*2 mm” should be written as “10×10×2 mm”. In page 5, line 2, “3*104” should be represented as “3×104”.
  3. In section “2.5.1. Animal model”, the strategy used to minimise potential confounders, such as animal/cage location, was not described. Please provide additional details.
  4. Figure 1C: add a label for the horizontal axis.
  5. Figure 8: the four sub-images within the figure 8 need to be labeled with letters.
  6. Page 20, line 13-28: the description of the H&E staining results for the organs is somewhat redundant. Please simplify or delete this part of the description.
  7. The methods section of the article mentions that the animals were sacrificed at 4 weeks and 8 weeks post-surgery. Also, both staining of important organs and micro-CT scans showed results at 4 weeks and 8 weeks. However, histological staining of mandible only showed results at 4 weeks. Please supplement the 8-week mandibular histology results and provide a description.
  8. Elaborate on the advantages of magnesium gallium alloy over other magnesium alloys as a potential barrier material for guided bone regeneration, and provide relevant literature support. Moreover, the following literatures on membrane design should be mentioned in discussion: PMID: 35224291; PMID: 36875054.
  9. The claim in page 26, line 2-6 lacks a supporting reference.
  10. In the immersion experiment, DMEM culture medium was used as the immersion medium. Why was it chosen instead of SBF, Hank's solution, etc.? Please add this to the discussion section.

Reviewer 2 Report

Comments and Suggestions for Authors
  1. terminology & Units: Consistently add "%" after numerical values in "Mg-xGa (x = 1, 2, 3 wt.%)"(e.g., 1%, 2%, 3%). 

          Add "mm" after all dimensions such as "10×10×2 mm" should be "10 mm × 10 mm × 2 mm". 

  1. What is the preparation method of Mg-Ga alloys? It should be clarified in Materials and Methods section.
  2. Fig. 2: Panels C and D appear to reuse identical data. Remove one to avoid redundancy.
  3. Fig. 3: The scale bar is illegible; please provide a higher-resolution image.
  4. Fig. 5: Label all unassigned XRD peaks (specifically for Mg-1Ga and Mg-2Ga).
  5. All instances of “in vitro” and “in vivo” must be formatted in italics throughout the text
  6. Carefully check superscripts and subscripts of the whole manuscript. 
  7. Page 11/Fig. 5 Discussion: While citing references stating corrosion products include MgO, carbonate, phosphate, and Ca-P salts, the authors only detected Mg(OH)â‚‚ in their data, and they failed to explain this discrepancy. Please provide a plausible explanation for this discrepancy
  8. Discussion Section: The discussion was not in-depth enough. While the experimental results are comprehensive, the discussion section remains predominantly descriptive rather than analytical. The current content focuses on experimental observations but lack mechanistic insights and critical analysis. This part needs major revision.
Comments on the Quality of English Language

The overall language is not concise enough. For example, "it can be seen that" and "As can be seen in the figure" are verbose phrases and can be deleted. And some sentences are too long, should be shortened (e.g., Page 8, Lines 9–13).  Please check through the text.

Reviewer 3 Report

Comments and Suggestions for Authors

The manuscript is well written, I thank the authors for their hard work, which I read with interest. 

  • There are statistical analyses of the results, but did the Authors make any 'minimum sample size analysis' (power analysis)? If available, please include the statistical analyses for the in vivo test and other relevant tests.
  • The software for the micro CT analysis was written incorrectly. NRecon is used for the reconstruction, and CTAn is generally used for analysis. This section should be controlled.
  • In the Discussion section, there are very limited citations, and some parts of the section sound like results. Please consider reorganizing.
  • Please check spelling and grammar throughout the manuscript, such as in vivo and in vitro.
  • The reference list includes two numbered lists
  • The references should be updated with more recent articles.

Reviewer 4 Report

Comments and Suggestions for Authors

This submission explores a novel approach using Mg-Ga alloys as biodegradable barrier membranes for guided bone regeneration (GBR). The utilisation of magnesium alloys is advantageous due to their biodegradability, which obviates the necessity for repeated surgical intervention. The utilisation of magnesium and gallium alloys is advantageous due to their enhanced mechanical properties. The relevance and importance of this study are indisputable.

Despite the fact that the results and conclusions presented were occasionally challenging to comprehend, the work is well structured and generally presented in a logical manner.

The authors have conducted a substantial amount of experimental work, encompassing both in vitro and in vivo experiments. In addition, they have investigated the biodegradation of magnesium alloys by measuring mass loss and the pH of the medium.

The results are discussed in detail, and the conclusions drawn are reasonable and supported by the results.

In summary, I would rate the manuscript very highly but advised the authors to make a few changes to improve the manuscript:

1) The authors did not sufficiently discuss the issue of cytotoxicity of Mg3Ga samples. Whether the cause of cytotoxicity is too high gallium content or too high biodegradation rate of the alloy. To answer this question it was necessary to investigate the degradation behavior of this alloy as well, but the authors did not do this. Is it possible to make any reasonable assumptions about the causes of cytotoxicity of Mg3Ga?

2) Abstract: “In this study, gallium (Ga) was innovatively introduced into Mg to enhance the mechanical strength and optimize the degradation behavior…”. I'm not sure if the authors first used these alloys, so I shouldn't use the term ‘innovatively’

3) Section 3.1. “The cell viability at day 7 in the Mg-3Ga group was below the cytotoxicity threshold (70%)…” What is this threshold, where is it taken from and please provide a reference?

4) Section 3.1. “the corrosion rate gradually decreased from 2.76 mm/y to 2.51 mm/y in 1-3 days and then increased to 2.56 mm/y in 7 days, and then gradually decreased thereafter “– In my opinion, the difference is very small to draw conclusions about the change in trend. Also, the presented discussion of corrosion mechanisms in this section requires more information (SEM microphotographs of the material surface, in situ analyses of the precipitates, etc.) or references to the literature

5) The text between fig 3 and 4. From the data presented in Fig. 4, the discussed difference in chlorine content is not obvious. Can you present the quantitative results. The conclusion in the last sentence (Combined with the elemental…) in this paragraph is also not obvious and requires either a more detailed explanation or a reference

6) The text between figures 4 and 5 says: "The crystalline phase Mg (OH)â‚‚ and the amorphous phase (the diffraction peaks are located at 15-30°)." You can't spot the amorphous phase in an XRD analysis. Please explain what you mean by 'amorphous phase'.

7) The results of the blood tests were strange. There was no difference in magnesium content, not only for the MgGa samples but also for the titanium meshes.

8) The size of some figures (1a, 7, 9, 12) should be made bigger. Their information is very difficult to read. I would also suggest improving Figure 4. It would be better to put the elemental mapping for each sample directly under the SEM images for that particular sample.

Reviewer 5 Report

Comments and Suggestions for Authors
  1. Vague Claims About Clinical Advantages

The manuscript states Mg-1Ga alloy offers a "promising solution for GBR applications" but fails to clearly differentiate its clinical superiority over existing titanium meshes or collagen membranes1. Phrases like "bridges the gap" and "optimized pore design" are overly general, lacking comparative data on mechanical performance, degradation timelines, or long-term outcomes in human-relevant models1. For instance, while residual strength post-degradation is reported (212 MPa at 28 days), no direct comparison is made to titanium mesh (~204 MPa). Such ambiguity undermines the claimed innovation.

  1. Unsubstantiated Resorption Claims The assertion that Mg-1Ga "will be completely resorbed" lacks evidence. The study only evaluates degradation over 28 days in vitro and 8 weeks in vivo—insufficient to confirm full resorption in humans1. Rabbit models cannot replicate human bone remodeling rates, and no data on Ga ion accumulation or long-term metabolic pathways are provided1. The statement "leaving no implant residue" is speculative and unsupported by multi-year follow-up.
  2. No Economic Feasibility Analysis

The authors neglect cost-benefit comparisons with existing barrier membranes (e.g., titanium mesh: $200–$500; collagen membranes: $100–$300)1. Mg-Ga alloy production costs, scalability, and surgical handling requirements are unaddressed. For clinical adoption, innovations must demonstrate either clear efficacy superiority or cost-effectiveness—neither is substantiated here.

Minor Flaws

  • Language Issues: The conclusion section contains redundant phrasing (e.g., "good biocompatibility, controlled degradation, and osteogenic properties") without quantifiable benchmarks1.
  • Overlooked Literature: No discussion of competing biodegradable metals (e.g., Mg-Zn alloys) or recent advances in polymer-ceramic composites for GBR1.
Comments on the Quality of English Language

Redundant phrasing (e.g., "good biocompatibility, controlled degradation") lacks quantifiable benchmarks

Round 2

Reviewer 2 Report

Comments and Suggestions for Authors

The authors have revised the manuscript properly.

Author Response

Thank you very much for your affirmation of this article and for your valuable suggestions for improvement.